# OpenReview forum: "Towards Understanding Safety Alignment: A Mechanistic Perspective from Safety Neurons"
_NeurIPS.cc/2025/Conference — NeurIPS 2025 poster_

### Official Review · Reviewer_d9S9 · 2025-06-18

**Clarity:** 3
**Significance:** 3
**Originality:** 3
**Rating:** 5
**Confidence:** 3

**Summary:**

The authors propose a method for identifying specific MLP neurons in the context of the safety alignment of large language models (LLMs). Specifically two complementary techniques are presented
- Inference-time activation contrasting. This technique identifies neurons by comparing their activation patterns between pre- and post-safety-finetuned model checkpoints.
- Dynamic activation patching. This technique employs causal interventions to quantify the extent to which the identified neurons are responsible for the model's safety behaviors.

These findings suggest a pathway toward more controlled and robust alignment of LLMs with human values and safety requirements.

**Questions:**

[Q1] Could you tell me the differences from the following existing work on task vector? What is the merit of focusing on neurons?

- Ilharco, Gabriel, et al. "Editing models with task arithmetic." arXiv preprint arXiv:2212.04089 (2022).
- Bhardwaj, Rishabh, Do Duc Anh, and Soujanya Poria. "Language models are Homer simpson! safety re-alignment of fine-tuned language models through task arithmetic." arXiv preprint arXiv:2402.11746 (2024).

**Ethical Concerns:**

["NO or VERY MINOR ethics concerns only"]

**Final Justification:**

This paper is technically solid and make a good contribution to AI/ML community. The rebuttals are satisfactory.

**Limitations:**

Limitations have been fully discussed in Appendix F.

**Quality:**

3

**Strengths And Weaknesses:**

### Strengths

- This paper is well-written organized. I think it is easy to follow.

- The motivations behind this paper is well-explained, and preliminaries are sufficiently presented before presenting the key techniques.

- The key findings may be very helpful for enhancing the safety of future LLMs. Given the growing importance of safety in AI, this paper can be impactful in AI community.

- The authors tested their method on a variety of model families and datasets. I appreciate the authors' attitude to show that the key findings are applicable not only specific experimental settings but as a general knowledge.

### Weaknesses

- While the supplementary material is submitted, the source code is empty (I'm sorry if this is an issue on my side). I am wondering whether I should trust the experimental results.

- In lines 331-343, while relevant studies are cited, I do not think that the related work is sufficiently discussed. I recommend discussing specifically what is essentially different, rather than discussing superficially.

---

> ### Author Rebuttal · Authors · 2025-07-30
>
> Thank you for your positive feedback and valuable suggestions. Please see our responses below.
>
> ## W1 The supplementary material is submitted, the source code is empty
>
> We sincerely apologize for the oversight. Upon reviewing the supplementary material, we find it is indeed empty. We are unsure at which stage this error occurred, but the outermost folder of our codebase is present, and the structure is indeed consistent with our codebase. Rest assured, we will make the entire codebase publicly available in the future.
>
> ## W2 Lack of sufficient discussion of related work
>
> Here is our discussion of related work. The discussion will be added to the paper if more space is permitted.
>
> Existing interpretability research on LLM safety can be broadly categorized into two perspectives: **Representation Engineering (RepE)** and **Mechanistic Interpretability (MI)**. We acknowledge the importance of RepE-focused studies, as they often demonstrate strong practical effectiveness in steering model behavior. For instance, [1-2] are firmly grounded in the RepE perspective, and [3][7] also incorporate some perspectives of this approach.
>
> In contrast, our work adopts the MI perspective, which seeks a bottom-up understanding of models’ inner workings. This perspective emphasizes the importance of localizing model functionality to the most fundamental operational units—a core principle of mechanistic interpretability. In the case of transformers, MLP neurons constitute approximately two-thirds of the model's parameters and serve as the foundational units for functionality. Therefore, we focus our study on neurons as the target of analysis to uncover safety mechanisms.
>
> For articles categorized under the MI perspective, [4] has been discussed in our paper, where we point out that toxicity is an incomplete part of model safety concerned in our work, a view also acknowledged in recent work [7]. [5] adopts a different definition of “neuron”, which describes individual parameters rather than complete functional units in this paper. Since features in transformers are usually represented as vectors, it is difficult to interpret how different parameters in a single vector play different mechanistic roles. [6] identified attention heads critical for LLMs’ safety. We believe that the functionalities of neurons and attention heads are not in conflict; instead, complex functions like safety are more likely to result from their collaboration. We plan to further explore their relationship in future work. [7] adopts a safety layer perspective, which we consider too coarse-grained compared to neurons and attention heads for providing a mechanistic understanding. [8] finds safety neurons based on their contribution to the residual stream of safety-related data. This ablation-based method may be influenced by the existence of the "Hydra effect", which suggests that ablating certain components in LLMs may trigger compensatory behaviors in other components.
>
> [1] Zou, Andy, et al. "Representation engineering: A top-down approach to AI transparency." arXiv 2023.
>
> [2] Arditi, Andy, et al. "Refusal in language models is mediated by a single direction." NeurIPS 2024.
>
> [3] Lee, Andrew, et al. "A Mechanistic Understanding of Alignment Algorithms: A Case Study on DPO and Toxicity." ICML 2024.
>
> [4] Yang, Yushi, et al. "Ablation is Not Enough to Emulate DPO: How Neuron Dynamics Drive Toxicity Reduction." MINT: Foundation Model Interventions. 2024.
>
> [5] Wei, Boyi, et al. "Assessing the Brittleness of Safety Alignment via Pruning and Low-Rank Modifications." ICML 2024.
>
> [6] Zhou, Zhenhong, et al. "On the Role of Attention Heads in Large Language Model Safety." ICLR 2025.
>
> [7] Li, Shen, et al. "Safety Layers in Aligned Large Language Models: The Key to LLM Security." ICLR 2025.
>
> [8] Zhao, Yiran, et al. "Understanding and enhancing safety mechanisms of LLMs via safety-specific neuron." ICLR 2025.
>
> ## Q1 Could you tell me the differences from the following existing work on task vector? What is the merit of focusing on neurons?
>
> Neurons are the fundamental computational units in neural networks, and they inherently possess interpretability attributes. For example, a neuron's "key" (a row in the MLP up-projection matrix) determines the type of input that will activate it, while its "value" (a row in the MLP down-projection matrix) defines its influence on downstream components, such as other neurons or attention heads. Through the residual stream, neurons can directly impact the output. This allows for a more granular explanation of the model’s internal mechanisms.
>
> In contrast, the task vector is in the entire model weight space. Although the task vector can steer the model’s behavior, it is a global feature of the model involving all the model parameters; understanding how it operates requires examining all the model activations, while the safety neurons identified in this paper are local.

---

> > ### Comment · Reviewer_d9S9 · 2025-08-04
> >
> > Thank you for the feedback. I am satisfied with the authors' responses. Please make sure to enrich the related work section (at least in the appendix). I will keep my original rating.

---

> > > ### Author Response · Authors · 2025-08-09
> > >
> > > We are pleased to have resolved your concerns. We will make sure to enrich the related work section and will also open-source our work.

---

### Official Review · Reviewer_nxdL · 2025-07-01

**Clarity:** 2
**Significance:** 3
**Originality:** 3
**Rating:** 3
**Confidence:** 3

**Summary:**

This paper investigates the mechanisms behind safety alignment in large language models by identifying "safety neurons" responsible for safe behaviors using mechanistic interpretability. The authors introduce inference-time activation contrasting to locate these neurons and dynamic activation patching to assess their "causal impact" on model safety. Experiments show that patching around 5% of identified neurons restores over 90% of safety performance on red-teaming benchmarks without compromising general ability.

**Questions:**

See cons.

**Ethical Concerns:**

["NO or VERY MINOR ethics concerns only"]

**Final Justification:**

Some of my concerns have been addressed, and so I decided to keep the original ratings.

**Limitations:**

See cons.

**Paper Formatting Concerns:**

No.

**Quality:**

2

**Strengths And Weaknesses:**

Pros:
- 1 Studying the mechanistic interpretability of LLMs' safety is indeed a very interesting and significant problem.
- 2 The proposed method that identifies the "safety neurons" seems reasonable
- 3 The effectiveness of the proposed method has been tested over multiple LLMs and Benchmark datasets. Also the method provides some understanding of "alignment tax", though not so rigorously tested.

Cons:

- 1  Some technical content is difficult to understand. How and why Dynamic Activation Patching works needs further explanation.  Eg.g., In what way does such a method link to the actual effect?
- 2 The Mechanistic Explanation in this paper seems insufficient.  While it successfully identifies which neurons are critical and reports their sparsity and so on, it fails to provide an explanation for how these neurons propagate a "safety signal" through the network, e.g., by checking its semantic meaning.
- 3 The proposed method seems analogous to traditional model compression techniques in keeping a few nodes/weights in the neuron networks. From my understanding, it is a kind of model compression and a bit exaggerated to claim this is related to the "safety" mechanism of LLM.

---

> ### Author Rebuttal · Authors · 2025-07-30
>
> Thank you for your questions and suggestions. Please find our responses below.
>
> ## W1 Some technical content is difficult to understand.
>
> In the process of generating each token, we use activation patching to modify the model’s forward pass, which results in responses that differ from those of the original model. We then assess the safety gap between the responses generated by the patched model (base/SFT) and the model used for patching (DPO). The smaller this gap, the more precisely the safety neurons capture the safety mechanisms. Thank you for your suggestion; we will include a more detailed explanation of the underlying intuition in the revised version.
>
> ## W2 The Mechanistic Explanation in this paper seems insufficient
>
> We have attempted to interpret the activations of these neurons by checking the semantic meaning of activating tokens (Table 2), like many previous works, but there is no clear pattern observed. This indicates that more in-depth interpretability techniques are required for interpreting safety mechanisms, which is a major challenge, and we leave it to future work. The contribution of this work lies in developing techniques for localizing safety mechanisms, which is the first step for interpreting them.
>
> ## W3 The proposed method seems analogous to traditional model compression techniques
> Indeed, our approach is similar to model compression, but there are two key differences:
> 1. **Unit of Focus**: Model compression typically targets individual parameters, which are not necessarily interpretable. In contrast, we focus on neurons, which have been shown to have specific, interpretable functions in LLMs.
> 2. **Intervention and Causal Effect**: Our method allows for direct intervention on identified neurons, demonstrating their active role in model inference. While neuron-level model compression exists, our comparisons (Lines 177-185) suggest that applying model compression for interpretability may not fully capture the causal effects.
>
> Additionally, beyond the technical differences, localizing neurons for interpretability is a well-established research direction, with distinct goals from model compression [1-3]. The research community generally recognizes these as **separate lines of study**.
>
> [1] Dai, Damai, et al. "Knowledge Neurons in Pretrained Transformers." ACL 2022.
>
> [2] Wang, Xiaozhi, et al. "Finding Skill Neurons in Pre-trained Transformer-based Language Models." EMNLP 2022.
>
> [3] Gurnee, Wes, et al. "Finding Neurons in a Haystack: Case Studies with Sparse Probing." TMLR.

---

### Official Review · Reviewer_4DAN · 2025-07-03

**Clarity:** 3
**Significance:** 2
**Originality:** 3
**Rating:** 4
**Confidence:** 4

**Summary:**

The paper investigates mechanistic roots of safety alignment in large language models by pinpointing a sparse set of “safety neurons” whose activations correlate with—and can causally induce—safe behaviour. It introduces a two-stage framework: inference-time activation contrasting ranks neurons whose activations differ between an unaligned and a safety-aligned model, and dynamic activation patching selectively substitutes those activations to verify that they alone can drive safer outputs.

**Questions:**

1. **Incremental Contribution** – What concrete contribution does this paper provide beyond prior neuron-steering work such as Representation Engineering [1], Sparse-Autoencoder Refusal Control [2], and Circuit Breakers [3]?
2. **Sparsity Stability** – The study reports that roughly 5 % of neurons govern safety. Does this fraction remain stable as model size scales (e.g., 2 B → 7 B → 70 B), or does the sparsity pattern shift?
3. **Robustness to Jailbreaks** – Do the identified safety neurons still trigger under stronger or more diverse jailbreak attacks (e.g., rewrite the harmful query in a benign way or use PAIR[4])?
4. **Activation Timing** – At which token positions are the safety neurons most strongly activated? Are they primarily triggered by system/chat-template tokens (e.g., `assistant\n`) before generation starts, or do they also fire in response to tokens within harmful prompts (and will the pattern be related to the length of the harmful queries)?
5. **Safeguard Rationale** – The safeguard is trained on HH-Harmless SFT activations yet evaluated on both SFT and DPO models that already refuse harmful queries when these neurons fire. What additional safety benefit does the safeguard offer in this setting?
6. **Practical Patching** – In a real deployment, we rarely know a priori which queries are harmful. How would the proposed patching procedure be used to make an *unaligned* base model safer?



[1] Zou, A., Phan, L., Chen, S., Campbell, J., Guo, P., Ren, R., ... & Hendrycks, D. (2023). Representation engineering: A top-down approach to ai transparency. arXiv preprint arXiv:2310.01405.

[2] O'Brien, K., Majercak, D., Fernandes, X., Edgar, R., Bullwinkel, B., Chen, J., ... & Poursabzi-Sangde, F. (2024). Steering language model refusal with sparse autoencoders. arXiv preprint arXiv:2411.11296.

[3] Zou, A., Phan, L., Wang, J., Duenas, D., Lin, M., Andriushchenko, M., ... & Hendrycks, D. (2024, June). Improving alignment and robustness with circuit breakers. In The Thirty-eighth Annual Conference on Neural Information Processing Systems.

[4] Chao, P., Robey, A., Dobriban, E., Hassani, H., Pappas, G. J., & Wong, E. (2025, April). Jailbreaking black box large language models in twenty queries. In 2025 IEEE Conference on Secure and Trustworthy Machine Learning (SaTML) (pp. 23-42). IEEE.

**Ethical Concerns:**

["NO or VERY MINOR ethics concerns only"]

**Final Justification:**

My concerns have been addressed, and I will update my score accordingly.

**Limitations:**

see questions

**Paper Formatting Concerns:**

no major format issue

**Quality:**

3

**Strengths And Weaknesses:**

#### **Strengths**

- Clear writing, diagrams make the paper easy to follow.
- Two-stage “activation contrasting => patching” pipeline is well motivated and causally validated.
- Provides a neuron-level explanation of the alignment tax

#### **Weaknesses**

- All evaluations are on ≤ 7 B models; scalability to larger frontier LLMs is untested.
- Technique builds on existing activation-patching ideas; novelty is incremental and overlaps with concurrent studies on nuerons[1,2].



[1] Zou, A., Phan, L., Chen, S., Campbell, J., Guo, P., Ren, R., ... & Hendrycks, D. (2023). Representation engineering: A top-down approach to ai transparency. arXiv preprint arXiv:2310.01405.

[2] O'Brien, K., Majercak, D., Fernandes, X., Edgar, R., Bullwinkel, B., Chen, J., ... & Poursabzi-Sangde, F. (2024). Steering language model refusal with sparse autoencoders. arXiv preprint arXiv:2411.11296.

---

> ### Author Rebuttal · Authors · 2025-07-30
>
> Thank you for your questions. We are happy to discuss these questions further, and below are our responses to them.
>
> ## W1 Scalability to larger frontier LLMs
> Due to time constraints, we are unable to conduct a full set of experiments during the rebuttal period. Below, we provide partial results for Llama2-13B (5\% safety neuron activation patching improves the safety of Base and SFT models, similar to the 7B model). Experiments with Qwen2.5-14B and Qwen2.5-32B are currently in progress, and we will include those results in a future revision.
>
> | Llama2-13 B            | BT    | RT    | GSM   | BBH   | MMLU  | TQA   |
> |------------------------|-------|-------|-------|-------|-------|-------|
> | Base                   | -4.5  | -4.0  | 0.22  | 0.151 | 0.507 | 0.268 |
> | Base*                  | -8.7  | -8.4  | 0.2   | 0.142 | 0.483 | 0.272 |
> | SFT                    | -7.5  | -5.8  | 0.165 | 0.133 | 0.525 | 0.268 |
> | SFT*                   | -11.2 | -10.3 | 0.165 | 0.132 | 0.528 | 0.278 |
> | DPO                    | -12.2 | -11.2 | 0.185 | 0.122 | 0.520 | 0.288 |
>
> ## W2 & Q1 Incremental novelty and contribution
>
> ### The novelty of the method builds on activation patching
>
> Activation patching is a technique used to examine the causal effects of specific components within a model. Previous works that employ activation patching typically identify components with causal effects through exhaustive enumeration. However, enumeration at the neuron level is infeasible due to the vast number of neurons. To address this gap, we propose inference-time activation contrasting, which first filters candidate neurons and then applies activation patching to identify safety neurons.
>
> ### About the prior neuron-steering work
>
> The primary goal of our work is not to provide a practical model steering method, but rather to serve as a first step toward understanding the safety mechanisms and safety alignment in large language models (LLMs). Therefore, we focus on decoding safety alignment into the most fundamental computational units—neurons.
> Representation Engineering [1] adopts a top-down approach, starting from high-level representations. While it is currently more practical, its contribution to interpretability is relatively limited. Circuit Breakers [2], as a follow-up work, shares a similar philosophy. Sparse Autoencoder (SAE) is a promising tool for interpretability, but whether SAE features are faithful to the original models remains an open question [3]. We also plan to investigate safety neurons from the perspective of SAEs in future work.
>
> [1] Zou, Andy, et al. "Representation engineering: A top-down approach to AI transparency." arXiv 2023.
>
> [2] Zou, Andy, et al. "Improving alignment and robustness with circuit breakers." NeurIPS 2024.
>
> [3] Leask, Patrick, et al. "Sparse autoencoders do not find canonical units of analysis." ICLR 2025
>
> ## Q2 Sparsity Stability
>
> From our current results, we observe that the sparsity of safety neurons is more closely related to the extent of model training rather than model size alone. Models trained during the same period—such as Llama-7B/14B, Mistral-7B, and Gemma-7B—exhibit similar levels of sparsity. In contrast, Qwen2.5-3B shows higher sparsity compared to these models (it has a similar number of neurons as Llama2-7B despite its smaller model size). This observation is also reflected in related work [4], which suggests that as model capabilities increase, the model's abilities become increasingly concentrated in a smaller, sparser subset of neurons. We think this is an interesting phenomenon for future research rather than a weakness of our work.
>
> [4] Cao, Yixin, et al. "Model Utility Law: Evaluating LLMs beyond Performance through Mechanism Interpretable Metric." arXiv 2025.
>
> ## Q3 Robustness to Jailbreaks
>
> To verify whether safety neurons faithfully capture the safety capabilities of aligned models, we conducted evaluations on diverse red-teaming datasets. These datasets include carefully crafted prompts, some of which are capable of successfully jailbreaking ChatGPT.
>
> ## Q4 Activation Timing
>
> We discuss this point in Appendix D.2, where we compute neuron importance using activations from different token positions: all tokens in the prompt, only the last token of the prompt, and all tokens in the generation. Our results show that using activations from the generation tokens yields the best performance. This suggests that safety neurons may play a more important role in influencing model behavior during output generation, while identifying harmful prompts may be more aligned with the function of attention heads[5].
>
> [5] Zhou, Zhenhong, et al. "On the Role of Attention Heads in Large Language Model Safety." ICLR 2025.
>
> ## Q5 Safeguard Rationale
>
> 1. We have not yet established whether the firing of safety neurons directly causes the model to refuse harmful queries; the relationship between the two may be more complex. However, our results indicate that the safeguard mechanism does indeed provide further improvements for DPO-aligned models.
> 2. In addition to enhancing safety, the safeguard mechanism can reject certain outputs before the model actually generates a response, thereby significantly improving inference efficiency. This point is also discussed in lines 313–316 of our paper.
>
> ## Q6 Practical Patching
>
> Our patching method is input-agnostic; that is, all experimental results in our paper are obtained by applying patching to all prompts, which does not need to identify harmful prompts beforehand.

---

> > ### Comment · Reviewer_4DAN · 2025-08-04
> >
> > Thank the authors for the detailed explanations and for providing additional experimental results. Including these new findings in the final version will greatly enhance the credibility and impact of the paper. My concerns have been addressed, and I will update my score accordingly.

---

> > > ### Author Response · Authors · 2025-08-09
> > >
> > > We are glad to have addressed your major concerns and truly appreciate your recognition of our work. We will include more comprehensive experimental results in the future version.

---

### Official Review · Reviewer_LPzo · 2025-07-06

**Clarity:** 3
**Significance:** 3
**Originality:** 4
**Rating:** 5
**Confidence:** 4

**Summary:**

The paper introduces a mechanistic interpretability framework to undestand how safety alignment emerges LLMs by identifying and analyzing “safety neurons” — a small, sparse subset of MLP neurons whose activations causally drive safer outputs without degrading general capability. Using inference-time activation contrasting to rank candidate neurons and dynamic activation patching to verify causality, the authors show that activating only these neurons can recover most of an aligned model’s safety performance across diverse red-teaming benchmarks while largely preserving LLM competence on different tasks. They further reveal that safety and helpfulness share many of the same key neurons but require different activation patterns, offering an explanation for the well-known alignment tax trade-off (aka helpfulness vs safety trade-offs). Finally, they found out that activation of the LLMs can be used as a safeguard that predicts unsafe generations.

**Questions:**

1. In lines 130-131 *“we choose the model after SFT as M1 (denoted as SFT) and the model after safety alignment as M2 (denoted as DPO).”* In my understanding, safety alignment is done in both SFT and DPO stage. The model without safety alignment is the base model. Lines 150-152 seem more correct.

**Ethical Concerns:**

["NO or VERY MINOR ethics concerns only"]

**Final Justification:**

All of my major concerns have been resolved. I think this is a good paper and I will keep my score.

**Limitations:**

yes

**Quality:**

3

**Strengths And Weaknesses:**

** Strengths **
1. Clear presentation and structure. The paper is straightforward to follow: the problem statement, methodological steps (identification, ranking and causal verification of safety neurons), and empirical validation are laid out in a logical flow, making the main take-aways easy to understand.


2. Well-motivated contribution. The work offers a fresh angle on how “helpfulness” and “safety” interact inside an LLM, complementing existing evaluations that treat alignment as a black-box phenomenon.


3. Insightful empirical findings. The observation that many key neurons are shared between helpfulness and safety, yet are activated in distinct patterns, sheds light on the long-discussed helpfulness-versus-safety trade-off.

** Weaknesses/Questions **

1. Generality beyond PEFT. All experiments fine-tune the base model with $(IA)^3$ adapters. It is unclear whether the same sparse “safety neuron” substructure—and the reported performance gap between helpfulness- and safety-oriented activations—would persist if one performed full-parameter SFT. A brief ablation on a fully fine-tuned checkpoint would clarify how much the conclusions depend on the PEFT setting.


2. Relation to steering-vector work. Prior studies such as [1] show that a single activation direction can steer the model away from refusal. How does the proposed neuron-level intervention compare with those steering approaches in terms of effectiveness, interpretability, and computational cost? More broadly, a short discussion situating this paper within the LLM-steering literature would help readers see the incremental novelty.


3. Predictive safeguard feels expected. If safety alignment trains the model to differentiate safe from harmful prompts, then using internal activations for prompt-level safety classification seems like a natural extension—similar to the logit-gradient test in [2]. Stating explicitly what new capability the proposed safeguard adds over existing gradient-based or embedding-based detectors would strengthen the practical significance claim.


[1] Refusal in Language Models Is Mediated by a Single Direction

[2] GradSafe: Detecting Jailbreak Prompts for LLMs via Safety-Critical Gradient Analysis

---

> ### Author Rebuttal · Authors · 2025-07-30
>
> Thank you for your recognition of our work and your valuable suggestions. Here are our responses to the weaknesses and questions.
>
> ## W1 Generality beyond PEFT
>
> We chose to employ $(IA)^3$ because full fine-tuning modifies the weights of neurons, making it challenging to ensure the neurons in the base model and fine-tuned model are consistent. We greatly appreciate your insightful suggestion about adding an ablation. Recent studies [1, 2] have indicated that full fine-tuning may only modify a small subnetwork of the model, suggesting that safety-related neurons might indeed be preserved during this process. As a result, we plan to investigate this aspect further in our future work.
>
> [1] Wu, Taiqiang, et al. "Shadow-FT: Tuning Instruct via Base." arXiv 2025.
>
> [2] Mukherjee, Sagnik, et al. "Reinforcement Learning Finetunes Small Subnetworks in Large Language Models." arXiv 2025.
>
> ## W2 Relation to steering-vector work
>
> Due to space limitations, we are currently unable to provide a comprehensive discussion on this matter in the paper. Our ultimate goal is to investigate the safety mechanisms of LLMs. Steering effectiveness, in this context, serves as a metric to assess the accuracy and reliability of the research objects we identify. Unlike papers focused on providing practical steering methods, this paper focuses on providing mechanistic interpretations at the neuron level. We believe there are intrinsic connections between neuron-level mechanisms and vector-steering, and we leave the detailed explorations for future work.
>
> ## W3 Predictive safeguard feels expected
>
> There may be two potential mechanisms through which safety neurons operate: one involves distinguishing harmful inputs, and the other increases the likelihood of rejection, or a combination of both. These mechanisms are not necessarily bound together. In other words, even if we can identify harmful inputs based on the model’s internal state, it does not guarantee that we can predict whether the model will reject the response. Our experimental results suggest that safety neurons may play a role in the latter mechanism, though the full mechanism remains to be explored.
>
> ## Q1 safety alignment is done in both SFT and DPO stage
>
> Our approach requires a pair of aligned and non-aligned models. However, we found that most open-source instruct models, either intentionally or unintentionally, have already undergone some form of safety alignment. As a result, we decided to start training from the base model. However, directly applying DPO safety alignment to the base model is not feasible. Therefore, we introduced a non-safety-aligned SFT phase as an intermediary step. The SFT stage here does not involve safety data but focuses on training basic instruction-following abilities to prepare safety alignment (DPO). We will add more details to the paper to make it clear.

---

> > ### Comment · Reviewer_LPzo · 2025-08-06
> > **Response after rebuttal**
> >
> > Thank you for you response. I will keep my score.

---

> > > ### Author Response · Authors · 2025-08-09
> > >
> > > Thank you for your feedback and for recognizing our work. We really appreciate your time and thoughtful comments.

---

### Decision · Program_Chairs · 2025-09-17

**Decision:**

Accept (poster)

**Comment:**

**(a) Summary**: The paper proposes a novel method that identifies the safety neurons of LLMs via analyzing causal associations between the safety behaviors of LLMs and the neuron activations. The empirical studies demonstrate several insights into the sparsity, transferability, and robustness of the identified safety neurons. The method also unveils that the helpfulness and safety have an inherent trade-off due to the large overlap between their causally-associated neurons. Lastly, the authors proposed to utilize safety neurons can be used to predict the unsafe generations inside the LLMs, serving as a post-alignment safeguard solution.

**(b) Strength**:
1. _Clear presentation and structure_. The paper is straightforward to follow from motivation to method, key findings, and applications in safeguarding.
2. _Well-motivated method_. The method is established upon a clear intuition that causal association can indicate the safety-important neurons.
3. _Insightful and practical empirical findings_. The proposed method helps explain the trade-off between helpfulness and safety. The findings also lead to practical mitigation of safety challenges.
4. _Comprehensive experiments_. Multiple reviewers appreciated the evaluation across different LLM families and red-teaming benchmarks, which adds robustness and generality to the claims.

**(c) Weakness**:
* _Insufficient mechanistic explanation_: While the paper successfully identifies "safety neurons" and demonstrates their causal influence, it does not fully explain how these neurons propagate safety signals within the network or what semantic meaning they might carry.
* _Discussion on the overlap with prior work_: Despite clarifications, the contribution still overlaps significantly with existing literature on representation engineering and steering vectors. The distinction between neuron-level interpretability and steering methods was explained, but the incremental novelty was not sufficiently emphasized or experimentally validated.
* _Limited large-scale validation_. The empirical studies (including rebuttal experiments) are limited to models with fewer than 30B parameters. But this is understandable for proof of concepts. An empirical study showing the scaling effects with the key findings will be helpful in the revision.

**(d) Reason for Acceptance**

Overall, the strengths of the paper outweigh the weaknesses.
1. Novel mechanistic method gaining insights into alignment: The paper introduces a neuron-level framework that identifies “safety neurons,” providing a fresh mechanistic explanation for the helpfulness vs. safety trade-off and advancing interpretability of LLM safety.
2. Strong empirical validation: Experiments across multiple LLM families and red-teaming benchmarks show that activating a small subset of neurons can recover most safety alignment benefits while preserving general model ability.
3. Impactful insights: Its findings not only deepen understanding but also open avenues for practical safeguards and alignment strategies.

Though concerns about the mechanistic explanation and analogy between the method and compression may remain as raised by Reviewer nxdL, the current merits (clear intuition and novelty) of the paper outweigh such limitations. I suggest that the authors clarify the limitations in the final revision.

**(e) Summary of Rebuttal**

During the rebuttal and subsequent discussions, the authors addressed most of the raised concerns:
* Generality beyond PEFT (Reviewer LPzo):
Authors explained that full fine-tuning modifies neurons, complicating consistency across models. They cited recent studies suggesting only small subnetworks change during fine-tuning and committed to testing this in future work. The reviewer accepted this reasoning and kept their score.
* Relation to Steering Work (Reviewers LPzo & 4DAN):
Authors clarified their focus is mechanistic interpretability at the neuron level, not practical steering methods. They emphasized that steering-vector and neuron-level approaches are complementary. This distinction satisfied reviewers, though they suggested a deeper discussion in the final version.
* Predictive Safeguard (Reviewer LPzo & 4DAN):
Authors explained that safety neurons may function not only by recognizing harmful inputs but also by modulating refusal likelihood. They further argued that the safeguard can improve inference efficiency by blocking unsafe generations early. Reviewers accepted this explanation.
* Scalability and Sparsity Stability (Reviewer 4DAN):
Authors provided new results on Llama2-13B, showing similar safety neuron sparsity patterns and safety improvements, with larger experiments (Qwen2.5-14B/32B) ongoing. They also argued that sparsity is tied more to training than size. Reviewer 4DAN appreciated the additional results, updated their score positively, and considered the concern addressed.
* Mechanistic Depth (Reviewer nxdL):
Authors acknowledged the limitation, noting no clear semantic patterns were found but framing their contribution as a first step in localizing safety mechanisms. They committed to adding further interpretability techniques in future work. Reviewer nxdL remained unconvinced about explanatory depth and kept a borderline-reject score.
* Analogy to Model Compression (Reviewer nxdL):
Authors clarified differences: their method intervenes at the neuron level with causal verification, unlike compression methods that target uninterpretable parameters. They also cited prior neuron-localization works to distinguish the contribution. Reviewer nxdL acknowledged the response but did not change their rating.
* Missing Code and Related Work (Reviewer d9S9):
Authors admitted the code upload error and promised public release. They also expanded their related work discussion, distinguishing between representation engineering (task-vector approaches) and mechanistic neuron-level interpretability. Reviewer d9S9 found this satisfactory and kept their positive rating.

**Final Decision Considerations**

Positive Weighing:
* Strong mechanistic interpretability contribution, with promising empirical results.
* Well-received by three reviewers (LPzo, 4DAN, d9S9) who maintained or increased supportive scores after rebuttal.
* New experimental results on larger models added credibility.
* Practical and theoretical significance in illuminating alignment tax mechanisms.

Negative Weighing:
* Reviewer nxdL maintained a borderline reject, citing insufficient mechanistic explanation and analogy to compression methods.
* Some novelty concerns remain, with overlapping territory relative to representation engineering and steering literature.
* Larger-scale validation and semantic interpretability are left as future work.

Overall, while there was one dissenting reviewer, the majority of reviewers found the rebuttal satisfactory, with strengthened support from two initially borderline reviewers. The paper’s contribution as a first step toward a mechanistic understanding of safety neurons was judged significant enough to warrant acceptance.